# Enamel Phenotypes: Genetic and Environmental Determinants

**DOI:** 10.3390/genes14030545

**Published:** 2023-02-22

**Authors:** John Timothy Wright

**Affiliations:** Adams School of Dentistry, University of North Carolina at Chapel Hill, Chapel Hill, NC 27599, USA; tim_wright@unc.edu

**Keywords:** enamel, development, phenotype, gene, epigenetic, environment, pathology

## Abstract

Dental enamel is a specialized tissue that has adapted over millions of years of evolution to enhance the survival of a variety of species. In humans, enamel evolved to form the exterior protective layer for the crown of the exposed tooth crown. Its unique composition, structure, physical properties and attachment to the underlying dentin tissue allow it to be a resilient, although not self-repairing, tissue. The process of enamel formation, known as amelogenesis, involves epithelial-derived cells called ameloblasts that secrete a unique extracellular matrix that influences the structure of the mineralizing enamel crystallites. There are over 115 known genetic conditions affecting amelogenesis that are associated with enamel phenotypes characterized by either a reduction of enamel amount and or mineralization. Amelogenesis involves many processes that are sensitive to perturbation and can be altered by numerous environmental stressors. Genetics, epigenetics, and environment factors can influence enamel formation and play a role in resistance/risk for developmental defects and the complex disease, dental caries. Understanding why and how enamel is affected and the enamel phenotypes seen clinically support diagnostics, prognosis prediction, and the selection of treatment approaches that are appropriate for the specific tissue defects (e.g., deficient amount, decreased mineral, reduced insulation and hypersensitivity). The current level of knowledge regarding the heritable enamel defects is sufficient to develop a new classification system and consensus nosology that effectively communicate the mode of inheritance, molecular defect/pathway, and the functional aberration and resulting enamel phenotype.

## 1. Introduction

Enamel, the hardest tissue in the body, covers the crown of human teeth and serves in a variety of capacities that are critical for normal tooth function and longevity [1]. Enamel has tremendous wear and fracture resistance due to its unique structure and composition. It can withstand the tremendous forces and pressures that are required for masticating the diverse foods consumed by humans [2]. Not only is this outer protective enamel layer extraordinarily resistant to wear and fracture, it also serves as an outstanding insulator against thermal and chemical challenges that occur during nutrient consumption [3]. This resilient outer covering of the dental crown serves to protect the tooth and allows it to function and be resistant to failure even when challenged by the rigors of mastication, chemical insults, thermal fluctuations, biofilm colonization and decades of use.

Enamel is derived from the oral epithelium that invaginates during tooth development [4]. Enamel has no capacity for repair or regeneration as the cells that form this tissue are lost during the developmental process and tooth eruption [1]. The process of enamel formation, amelogenesis, can be perturbed by a variety of genetic and environmental influences [1]. Phenotypic variation in human enamel is common and has been the subject of numerous investigations. The purpose of this review is to briefly describe the major events related to normal enamel development and then present our current understanding of the genetic and environmental determinants associated with abnormal enamel formation and the diversity of enamel phenotypes seen clinically.

## 2. Normal Enamel Development

Tooth formation in humans begins around the 8th week of gestation with the invagination of the oral epithelium that interacts with the underlying ectomesenchyme cells to initiate tooth bud development [4]. These two tissues interact through molecular signals including growth factors that direct the epithelial component to transform into the future cells that will give rise to the enamel-forming cells, the ameloblasts [5]. The forming tooth germs progress through a series of developmental stages that have been named for key events. These have been termed: (1) initiation—stage of initial tissue invagination and proliferation, (2) morphogenesis—development of three-dimensional crown form and differentiated cells, (3) matrix secretion—formation of extracellular matrix, (4) eruption—the tooth root elongates, and the crown emerges into the oral cavity [6]. Tooth formation can be modified or terminated at any of these developmental stages by genetic alterations and/or environmental stressors. There are numerous excellent reviews that describe the normal processes of tooth development and their molecular signaling and regulation [7,8].

The enamel organ, which is responsible for the formation of enamel, contains several epithelium-derived cell types that contribute to the development of the dental crown and enamel [9]. The inner and outer enamel epithelium are generated at the cervical loop, which serves as a dental epithelial stem cell niche for the developing tooth. The inner enamel epithelium gives rise to the ameloblasts [9]. The ameloblasts are initially adjacent to the mesenchymal tissue and odontoblasts that form the dentin. Epithelial-derived cells known as the stratum intermedium form a single cell layer on the non-secretory side of the ameloblasts. The stratum intermedium cells appear to be critical for normal enamel development, including stabilization of the ameloblast layer, although their role appears to be more complex than is currently [10] understood. The preameloblasts change their morphology as they differentiate into tall columnar cells that will secrete a unique extracellular matrix and then degrade and remove most of this matrix while creating a microenvironment necessary to allow for mineralization of the enamel [1,11]. Once the full thickness of the enamel has been laid down, the ameloblasts continue to degrade and remove the extracellular matrix in an orderly fashion by secreting specific proteinases (MMP20, KLK4) [11]. Removal of the extracellular matrix provides room for crystallite growth, allowing the enamel to achieve its highly mineralized state of about 96% mineral by weight [12,13]. The mineral component of enamel is carbonate-substituted hydroxyapatite that contains numerous trace elements, some of which, such as fluoride, can alter the enamel solubility and structure of the crystallites [14]. Enamel has a unique structure that varies between species and reflects the evolutionary development and modification in response to functional needs placed on the dentition [15].

Given the complexity of human enamel and the temporal and spatial requirements to develop such a refined tissue, it is not surprising that there are numerous known etiologies leading to abnormal enamel development. These alterations can occur due to genetic and epigenetic influences and can result from environmental exposures and stressors [16,17]. Conditions altering enamel development can result in changes in the amount, composition and or structure of the tissue (Figure 1) [16,18]. The enamel phenotypes (Figure 1) are diverse, with a decreased amount (enamel hypoplasia) and reduced mineral content (enamel hypomineralization) being the predominate malformations observed. Non-pathological variants, such as SNPS in genes coding for proteins that contribute to enamel formation (e.g., *AMELX*), have been associated with dental caries, presumably due to subtle changes in the enamel structure and composition [19,20,21]. Developmental defects of enamel (DDE) are common in the general population (up to 80%) and vary markedly in phenotype with a variety of hypoplastic and hypomineralized defects [22,23,24]. DDE are associated with pathologies and altered disease risk, including dental caries and enamel fracturing [25,26]. Morbidities including dental hypersensitivity, altered esthetics, dental caries, and enamel or even tooth loss are associated with DDE [27,28]. DDE also can be associated with co-morbidities such as abnormal tooth eruption, dental cysts, and dental malocclusions as well as with systemic manifestations and syndromes [28,29].

Animal studies have added greatly to our knowledge of both normal and pathological enamel formation. The advent of transgenic animals has allowed for detailed studies of gene expression, protein function, pathogenic mechanisms, and detailed phenotyping that would not be possible in humans [30,31,32]. For example, many studies assessed tooth enamel genes (e.g., *AmelX*, *Enam*, *Mmp20*) and different mutations, the resulting proteins, and how and why these changes were associated with different enamel phenotypes [33,34,35]. Studies of the mouse and human transcriptomes of ameloblasts have demonstrated the multitude and diversity of genes involved in amelogenesis [36,37]. While animal studies have added greatly to our knowledge and understanding of enamel defects and their etiologies, it is beyond the scope of this manuscript to evaluate the many different animal and molecular models described in the literature. The following sections provide an overview of developmental defects of human enamel and our current knowledge of their etiologies and phenotypes.

## 3. Genetically Determined Enamel Defects

Enamel formation is exquisitely controlled and highly regulated at the molecular level with over 10,000 genes being expressed by amelobasts during amelogenesis [37]. Additionally, there are hundreds if not more microRNAs that appear to help regulate gene expression and that appear to be important for normal enamel formation [38,39]. Some of the gene products are unique to enamel, but most genes and regulator elements involved in enamel formation are also functional in cells other than ameloblasts [37]. The *AMELX* gene that codes for the most prevalent enamel matrix protein, amelogenin, is thought to function only in enamel, although some investigators suggest there could be a function beyond enamel formation [40]. There are many examples of genes associated with enamel defects that are also causative of pathology in other tissues such as skin and hair [41]. Historically, hereditary enamel defects have been separated into syndromic and non-syndromic conditions [42,43]. Classification of the non-syndromic hereditary enamel defects were named using the nosology of the amelogenesis imperfectas (AI) [43]. Syndrome-associated enamel defects were not referred to as AI using the Carl Witkop Jr. nosology [42]. The AI conditions were subdivided based on mode of inheritance, phenotype, and perceived development mechanism (i.e., hypoplastic, hypomaturation, hypocalcified). This nosology for enamel defects developed by Carl Witkop Jr. in the 1950s and 1960s and refined in the late 1980s was remarkably insightful given that it was developed prior to knowledge of any of the AI-associated genes and remains in use to this day [42,43]. Revision of the AI nosology using our current knowledge of genes and molecular pathways has been proposed; however, a system has yet to be fully developed and adopted for general use [44,45].

Enamel phenotypes can be subtle and variable (as can systemic phenotypes), making the classification of hereditary enamel defects into syndromic and non-syndromic challenging, and terminology is used inconsistently in the literature [27,46]. In this overview, conditions are categorized using Online Mendelian Inheritance in Man designations along with genotype and phenotype information that has been augmented by searches in the literature to help clarify descriptions of phenotypes and molecular etiology.

Genes known to be associated with hereditary enamel defects code for proteins with diverse functions such as transcription factors, growth factors, extracellular matrix proteins, ion channels, cell structure and cell motility proteins [16]. The physiology of the ameloblasts and surrounding cells is regulated by genes that are critical for normal cellular function as well as those required for the unique processes of amelogenesis. For example, ameloblasts express genes coding for cellular connections and channels for shuttling ions and water, which help create the microenvironment necessary for enamel mineralization [36]. Given the diversity of proteins and functions necessary for the creation of enamel, it is not surprising that there are myriad genetic causes of enamel defects.

### 3.1. Syndrome-Associated Developmental Defects of Enamel

Genetic mutations causing changes in proteins that negatively impact normal enamel formation are often also causative of phenotypes affecting tissues beyond enamel [16,47,48]. This is not surprising given the ubiquitous expression of many genes and repurposing of proteins for different functions in different cells and tissues. These genetically determined developmental defects of enamel (DDE) are thus designated as syndrome-associated hereditary enamel traits. The types of genes involved and the dysfunction of the encoded proteins result in a diverse spectrum of systemic and enamel phenotypes. A decreased amount or hypoplastic enamel (e.g., thin, pitted, grooved) is the most common syndrome-associated enamel trait; however, hypomineralized enamel also occurs. Table 1 presents syndromes with enamel phenotypes and a known molecular basis, which were cataloged in the Online Mendelian in Man database (final search: 29 December 2022) [49]. The search revealed 95 syndromes with a reported enamel phenotype (30 autosomal dominant, 60 autosomal recessive, 5 X-linked). Some genes associated with syndromes can also result in a phenotype that involves only enamel. For example, the *LAM3* gene produces a protein important for normal laminin-5 formation and has been shown to be causative of autosomal dominant hypoplastic AI (OMIM# 104530) as well as junctional epidermolysis bullosa (OMIM#s 226650, 226700) [49]. The group of conditions known as epidermolysis bullosa are associated with blistering of the skin and mucous membranes due to weakening of cellular attachments such as that resulting from abnormal LAMININ–5. Other genes are listed in OMIM as being causative of both syndromic and non-syndromic enamel traits (e.g., *DLX3*, *FAM20A*) [49]. There are numerous syndromes listed using the term AI to describe the enamel phenotype (e.g., *PEX1, TMEM165, MTX2, DNAJC21, SLC13A5, SLC10A7, ROGDI*) (Table 1). Genetic mutations causing syndromes with enamel phenotypes encode proteins that are extremely diverse in their functions, including transcription factors, extracellular matrix proteins, cell signaling, growth factors, enzymes, cell adhesion, and organelle function to name a few [16].

### 3.2. Amelogenesis Imperfectas; Non-Syndromic Hereditary Enamel Defects

The number of AIs with an identified molecular etiology have grown markedly over the past three decades following the identification of *AMELX* mutations in 1991 [50]. The OMIM database lists 20 associated conditions falling under this phenotype series and uses an alphanumeric designation that is based on the Witkop classification using phenotype for the major groupings (i.e., I—hypoplastic, II—hypomaturation, III—hypocalcification, IV—hypomaturation/hypoplastic with taurodontism) [49]. Table 2 lists 18 of these conditions with their mode of inheritance, associated gene, and phenotype. The phenotypes are described as hypoplastic or hypomineralization, as designations of hypomaturation and hypocalcification are both purported developmental mechanisms leading to a hypomineralized phenotype. Two conditions listed in OMIM are included in Table 1 as syndrome-associated enamel defects as the published cases of *DLX3*-associated amelogenesis imperfecta have an attenuated phenotype of Tricho-dento-osseous syndrome (OMIM# 190320), and the *FAM20A* gene is associated with renal calcifications and is listed using the nosology of enamel–renal syndrome (OMIM # 204690) [51,52].

The only X-linked AI gene known is *AMELX*, which codes for the most abundant enamel matrix protein amelogenin. The *AMELX* phenotypes vary substantially from generalized thin hypoplastic enamel to hypomineralized with a normal or near normal enamel thickness and combinations of the two [53]. Different allelic *AMELX* mutations result in a variety of changes across the AMELX protein that result in enamel hypoplasia and/or hypomineralization defects, depending on the nature of the protein alteration and domains [53]. Although *AMELX* (OMIM# 301200)-associated AI is listed in OMIM as X-linked dominant, it shows a classic phenotype of an X-linked recessive trait and was used to exemplify lyonization and the variable phenotype seen between males and females [54,55]. The six known autosomal-dominant AI traits involve a variety of extracellular matrix genes including *ENAM, LAM3,* and *AMTN* [45]. The *SP6* gene codes for a transcription factor and the function of FAM83H, while still being obscure, and it could be involved in endoplasmic reticulum-to-Golgi vesicle trafficking and protein secretion [56,57].

The ten known autosomal recessive AI-associated genes are diverse in their purported functions. The two major proteinases, MMP20 and KLK4, are critical for processing the extracellular matrix during the secretory and maturation stages [58]. Tryptophan-aspartate repeat domain 72 (WDR72) has recently been posited to have functions associated with directing the microtubule assembly necessary for membrane mobilization and subsequent vesicle transport [59]. The ODAPH protein’s (coding gene also called *C4orf26*) phosphorylated C terminus of C4ORF26 promoted hydroxyapatite nucleation and supported crystal growth; however, the specific pathogenesis leading to the hypomineralized phenotype is not well understood [31,60]. The *SLC24A4* gene encodes a calcium transporter (protein is *NCKX1)* that is upregulated in ameloblasts during the maturation stage of amelogenesis. There are over 300 known solute carrier (SLC) transporters that move substances across biological membranes [61]. Mutations in two of these genes are associated with syndromic enamel defects (*SLC13A5, SLC10A7)*, and one appears to have only an enamel phenotype (*SLC24A4*). The *RELT* gene encodes for a member of the tumor necrosis factor receptor super-family and is expressed in the secretory stage of amelogenesis with mutations being associated with a hypomineralized phenotype [62]. Integrins, such as the one encoded by the *ITGB6* gene, are cell surface glycoproteins that function in cell–cell and cell–matrix adhesion. Mutations in *ITGB6* cause enamel hypoplasia with no other apparent phenotype shown in humans [63]. There are numerous proteins that involve cell–cell and cell–matrix adhesion that are associated with enamel defects (typically enamel hypoplasia) including the ITGB4 (See Table 1) [64]. *ACP4* belongs to the histidine phosphatase superfamily and is expressed by the secretory-stage ameloblasts but not by maturation-stage ameloblasts [65]. It is thought that APC4 functions in processing and regulating enamel matrix proteins at the mineralization form at the distal ends of the secretory ameloblasts [66]. The gene encodes a G protein-coupled receptor that functions as a proton-sensing receptor that has pH sensitive activity [67]. Mutations in GPR68 are associated with a hypomineralized phenotype [67]. Other G protein-coupled receptors are critical for enamel formation. Although not yet associated with human pathology, *Gpr115* is expressed by ameloblasts in maturation-stage ameloblasts and is indispensable for the expression of carbonic anhydrase 6 (*Car6*) [68]. Given the complexity of enamel formation that involves molecular regulation of numerous pathways and developmental mechanisms, there are sure to be numerous additional genetic mutations that will be associated with DDE.

## 4. Environmental Etiologies of Developmental Defects of Enamel

Developmental enamel defects that result from environmental causes are numerous in etiology and are diverse in phenotype. Given the timing, orchestration, and regulation of enamel formation, it is not surprising that the amount, composition, and structure of enamel can be perturbed by environmental stressors [69]. The teeth involved and the extent and characteristics of DDE are influenced by the timing of the event in relationship to the developmental stage of enamel, the duration, magnitude and type of stressor (see Table 3). Opacities and hypomineralization defects of the enamel are seen most often with enamel hypoplasia being less common based on epidemiological studies [22,69]. Excellent reviews on the topic of environmental etiologies of enamel defects have been published [17,70,71]. A recent scoping review identified 114 factors associated with DDE [17]. Factors associated with DDE can be clustered or grouped according to timing of the environmental exposure (i.e., prenatal, neonatal, postnatal) and by the type of stressor or mechanism of cellular insult (i.e., metabolism, blood supply, hypoxia, immune response, xenobiotic) [17,72]. One of the most studied and most common DDEs is dental fluorosis that results from ingesting excessive fluoride during the time of enamel formation. The prevalence of dental fluorosis varies in populations around the globe with very mild and above fluorosis reportedly affecting about 60% of the population in the United States [73]. Fluoride exposure can affect cellular physiology and the enamel mineral composition and structure in a dose-dependent manner [74,75,76,77]. The enamel phenotype varies from mild opacities to severe hypomineralization that results in enamel discoloration, fracturing, and loss [78,79]. While the cause of dental fluorosis is due to consumption and systemic exposure to fluoride, the exact mechanisms leading to the enamel defect and phenotype continue to be investigated. The contribution of genetics to the risk and resistance for developing dental fluorosis was observed in animals and has subsequently been increasingly studied in humans [80]. This will be explored further in the following section on gene interactions with environmental exposures.

Molar hypomineralization (MH), commonly referred to as molar–incisor hypomineralization, is a DDE associated with a variety of environmental etiologies including childhood illnesses that occurs between birth and 3 years of age [81,82,83]. Maternal illness is associated with MH, indicating that prenatal stressors can play a role as well as perinatal and postnatal factors such as prematurity, caesarean section birth, kidney disease, urinary tract infection, and gastric disorders, to name a few [84]. The prevalence varies between populations with a worldwide average prevalence of 13% [85,86,87]. The enamel phenotype varies from mild (enamel opacity and discoloration) to severe (marked hypomineralization with enamel loss upon tooth eruption) [88,89]. Hypomineralization of the primary second molars is associated with hypomineralization of the first permanent molars [90]. It has been suggested in a number of studies that the etiology and variability of phenotype associated with MH may be influenced by a variety of genetic factors [91,92,93]. Given the broad influence of genetics and genetic variation in the human genome and gene environment interactions, it is becoming generally accepted that MH is a complex condition resulting from a combination of environmental and genetic factors [94]. Gene–environment interactions such as those involved in conditions such as fluorosis and MH will be further reviewed in the following section of enamel development and epigenetics.

## 5. Gene and Environment Interactions Influencing Enamel Development

It is now believed that most human phenotypes have both a genetic and environmental contribution, although our understanding of those contributions and the genetics-to-phenotypes relationship is far from complete [95]. The traits that we observe clinically are the result of a variety of interactions and mechanisms resulting from gene–gene and gene–environment interactions (G x E). For some developmental defects in humans, the relationships of G x E are much clearer than they are for tooth development and enamel formation. The mechanisms for gene–gene interactions and G X E that can modify development are diverse and include environmental exposure of a substance with specific proteins, disruption of signaling pathways, epigenetics and microRNA modification [96]. How much contribution is genetic versus environment varies from one phenotype to another and has been investigated using a variety of methodological approaches. For example, there is 100% concordance for fetal alcohol syndrome in monozygotic twins compared with 64% in dizygotic twins, indicating genetic modulation of this clinical outcome [97]. Twin studies assessing enamel defects show varying results as to the contribution of heritability to the enamel phenotype [98,99,100]. Two twin studies assessing enamel defects in the primary dentition show conflicting results as to twin concordance [98,99]. A study of MH involving the first permanent molars suggested greater concordance of the trait in twins but also greater risk associated with environmental determinants, including family income level and gestational hemorrhage [100]. The influence of genetic variation has been investigated for several DDE including fluorosis and MH.

Studies of skeletal fluorosis indicate that fluoride increases osteoblast activity and increases expression of Wnt3A in a rat model, suggesting that fluoride affects the Wnt-βcatenin signaling pathway [101]. Other molecular mechanisms influenced by fluoride and potentially contributing to skeletal fluorosis include Hedgehog, Notch, parathyroid hormone, endoplasmic reticulum stress and epigenetics [101]. DNA methylation of the P16 gene promoter in a fluorosis model resulted in decreased P16 protein expression that is important in cell cycle regulation [101]. Fluoride can cause hypermethylation of *BMP1, MMP11* and other genes based on studies in a human osteosarcoma cell model [102]. Ameloblast cell models show that fluoride exposure influences calcium signaling pathways, apoptosis and causes endoplasmic reticulum stress [103,104]. While there are multiple studies examining SNPs in genes associated with fluorosis, studies examining the epigenetic contributions are limited [105]. One study of humans in a region with endemic coal-burning associate fluorosis-found hypermethylation rates of the O^6^–methylguanine–DNA–methyltransferase gene, a DNA repair gene affecting the severity of fluorosis [106].

There are few studies evaluating epigenetics and DNA methylation in relation to dental caries and hypomineralization of teeth. One twin study, suggesting multiple genes known to be associated with tooth formation, showed differential methylation in children with dental caries and hypomineralized second primary molars [107]. A study using cheek cells found no difference in global methylation of DNA in children with MH [108]. While these epigenetic studies are quite preliminary, collectively, they suggest the potential contribution of epigenetics to DDE either as an association, or possibly mechanistically, by altering the expression of genes important to normal enamel development. Examination of the role of bisphenol A exposure and the formation of enamel defects suggest that it may affect multiple biological processes, including having an epigenetic effect [109].

MicroRNAs are known to play an important role in regulating gene expression during enamel formation and DDE [110]. Evaluation of the role of microRNAs in gene regulation related to AI indicates that miR-16-5p and miR-27b-3p are likely involved through dysregulation of mouse AI genes [111]. Are the MicroRNAs involved in tooth formation influenced by environmental exposures as another contributor to DDE variance? This is an interesting question. One study assessing microRNA expression by rat fetuses having prenatal fluoride exposure shows differential expression of multiple microRNAs known to be involved in tooth formation and fluorosis pathogenesis [111]. Differential microRNA expression in the fluoride-exposed enamel organ was associated with functional annotation of target genes identified with pathways including the calcium signaling pathway and MAPK signaling pathway [111].

## 6. Conclusions

Our understanding of the complex genetic and environmental factors contributing to the diverse enamel phenotypes associated with DDE has advanced tremendously over the past 20 years. The known genes causing AI and syndromes with enamel phenotypes have grown dramatically in just the past decade. While our knowledge of the genetics and important environmental influences on development continues to grow, there are still many gaps in our knowledge, and the task of addressing these issues in human research models is challenging. The importance of epigenetics in tooth formation has been known for over four decades, but the first manuscript on MicroRNA expression and their role in tooth formation was published in 2008 [112,113]. The advancement of molecular biological tools and bioinformatics provides new opportunities to evaluate questions surrounding DDE and the mechanistic determinants and advance our knowledge surrounding the science of enamel formation. Our knowledge of the genetics causing DDE has advanced to the point that elaborating a more accurate and informative nosology for these conditions should be pursued to improve communication between patients, families, clinicians and researchers.

## Figures and Tables

**Figure 1 genes-14-00545-f001:**
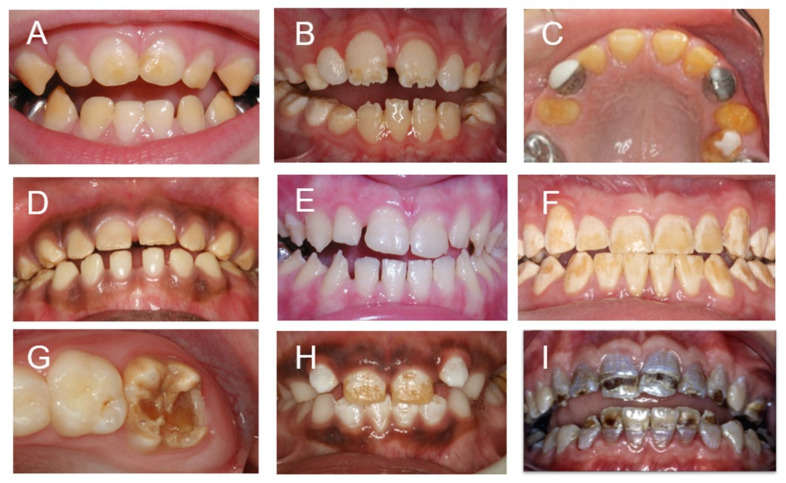
This panel illustrates the variable hypoplastic and hypomineralized phenotypes that can result from genetic and environmental etiologies: (**A**–**C**) syndrome-associated DDE; (**A**) hypomineralized enamel—#226750. KOHLSCHUTTER–TONZ SYNDROME; (**B**) hypoplastic enamel—#277440. VITAMIN D-DEPENDENT RICKETS; (**C**) hypoplastic enamel—#601216 DENTAL ANOMALIES AND SHORT STATURE; (**D**–**F**) are amelogenesis imperfectas; (**D**) hypomineralized enamel—#204700. AMELOGENESIS IMPERFECTA, HYPOMATURATION TYPE; (**E**) hypoplastic enamel—#104500. AMELOGENESIS IMPERFECTA, TYPE IB; (**F**) hypomineralized enamel—# 301200. AMELOGENESIS IMPERFECTA, TYPE IE; (**G**–**I**) environmental associated DDE; (**G**) hypomineralized enamel—MOLAR HYPOMINERALIZTION; (**H**) hypomineralized enamel—DENTAL FLUOROSIS; (**I**) combined hypoplastic hypomineralized enamel—KIDNEY DYSFUNTION and TETRACYCLINE.

**Table 1 genes-14-00545-t001:** Syndromes with known molecular etiology and enamel phenotype.

OMIM Syndromes with Enamel Trait	Gene	Inheritance	Phenotype
**Autosomal Dominant Conditions**			
#112240. COLE–CARPENTER SYNDROME 1; CLCRP1	*P4HB*	Autosomal Dominant	enamel hypoplasia, small teeth, growth failure, short stature, osteopenia
#119600. CLEIDOCRANIAL DYSPLASIA; CCD	*RUNX2*	Autosomal Dominant	enamel hypoplasia, supernumerary teeth, failure of exfoliation of the primary dentition, malocclusion
#125500. DENTINOGENESIS IMPERFECTA, SHIELDS TYPE III	*DSPP*	Autosomal Dominant	enamel hypoplasia/aplasia and pitting–some cases
#146300. HYPOPHOSPHATASIA, ADULT	*ALPL*	Autosomal Dominant	enamel hypoplasia
#149730. LACRIMOAURICULODENTODIGITAL SYNDROME; LADD	*FGFR2; FGFR3, FGFR10*	Autosomal Dominant	enamel hypoplasia, hypodontia, nasolacrimal duct obstruction, hearing loss
#151050. LENZ–MAJEWSKI HYPEROSTOTIC DWARFISM	*PTDSS1*	Autosomal Dominant	dysplastic enamel, growth abnormalities, sparse hair, micrognathia, hearing loss, delayed bone age
#162200. NEUROFIBROMATOSIS, TYPE I; NF1	*NF1*	Autosomal Dominant	enamel hypoplasia
#164200. OCULODENTODIGITAL DYSPLASIA; ODDD	*GJA1*	Autosomal Dominant	enamel hypoplasia, microcephaly, cleft lip/palate, CNS involvement
#166750. OTODENTAL DYSPLASIA	*11q13 microdel*	Autosomal Dominant	enamel hypoplasia, coloboma, hearing loss
#170390. ANDERSEN CARDIODYSRHYTHMIC PERIODIC PARALYSIS	*KCNJ2*	Autosomal Dominant	enamel hypoplasia, microcephaly, hypokalemic periodic paralysis
#173650. KINDLER SYNDROME	*KIND1*	Autosomal Dominant	enamel hypoplasia/dental caries
#180500. AXENFELD-RIEGER SYNDROME, TYPE 1; RIEG1	*PITX2; FOXC1*	Autosomal Dominant	severe enamel hypoplasia, conical and misshapen teeth, hypodontia, and impactions
#180849. RUBINSTEIN–TAYBI SYNDROME; RSTS	*CREBBP*	Autosomal Dominant	enamel hypoplasia, discoloration and wear, talon cusp, CNS involvement, short stature
#182290. SMITH–MAGENIS SYNDROME; SMS	*17p11.2; RAI1*	Autosomal Dominant	enamel hypoplasia/hypomineralized enamel (variable), brachycephaly, CNS involvement
#190320. TRICHODENTOOSSEOUS SYNDROME	*DLX3*	Autosomal Dominant	thin/pitted enamel, small teeth, taurodontism-marked variability, dense bone, kinky hair
#191100. TUBEROUS SCLEROSIS 1; TSC1	*TSC1*	Autosomal Dominant	enamel hypoplasia, pitted enamel, facial angiofibroma, seizures, tumorigenesis
#601005. TIMOTHY SYNDROME; TS	*CACNA1C*	Autosomal Dominant	enamel hypoplasia, cardiac defects, CNS involvement
#612462. PSEUDOHYPOPARATHYROIDISM, TYPE IC; PHP1C	*GNAS*	Autosomal Dominant	enamel hypoplasia, delayed tooth eruption, short stature, osteoporosis, CNS involvement
#612463. PSEUDOPSEUDOHYPOPARATHYROIDISM; PPHP	*GNAS*	Autosomal Dominant	enamel hypoplasia, delayed tooth eruption, short stature, osteoporosis, cataract
#613254. TUBEROUS SCLEROSIS 2; TSC2	*TSC2, IFNG*	Autosomal Dominant	enamel hypoplasia, pitted enamel, facial angiofibroma, seizures, tumorigenesis
# 615821. CARDIOMYOPATHY, DILATED, WITH WOOLLY HAIR, KERATODERMA, AND TOOTH AGENESIS; DCWHKTA	*DSP*	Autosomal Dominant	enamel hypomineralization, wooly hair, Palmoplantar keratoderma. cardiomyopathy
#617915. HYPOTONIA, ATAXIA, DEVELOPMENTAL DELAY, AND TOOTH ENAMEL DEFECT SYNDROME; HADDTS	*CTBP1*	Autosomal Dominant	hypomineralized discolored enamel, hypotonia, frontal bossing, ataxic gait
# 618458. KNOBLOCH SYNDROME 2; KNO2	*PAK2*	Autosomal Dominant	enamel hypoplasia primary dentition, eye abnormalities
#618874. CHROMOSOME 17q11.2 DUPLICATION SYNDROME	*17q11.2; 1.4-Mb*	Autosomal Dominant	enamel hypoplasia, microcephaly, premature baldness, seizures
#619980. BRADDOCK–CAREY SYNDROME 1; BRDCS1	*21q22 del*	Autosomal Dominant	enamel hypoplasia, glossoptosis, cleft palate, hypotonia
#619228. DEVELOPMENTAL DELAY WITH DYSMORPHIC FACIES AND DENTAL ANOMALIES; DEFDA	*SATB1*	Autosomal Dominant	enamel dysplasia, small teeth, CNS involvement
# 619229. KOHLSCHUTTER–TONZ SYNDROME-LIKE; KTZS	*SATB1*	Autosomal Dominant	enamel hypomineralization, delayed tooth eruption, seizures, CNS involvement
**Autosomal Recessive Conditions**			
#204690. ENAMEL–RENAL SYNDROME; ERS	*FAM20A*	Autosomal Recessive	Generalized hypoplastic and failure of tooth eruption, gingival hypertrophy, renal calcifications, also listed as OMIM #614253
#210600. SECKEL SYNDROME 1; SCKL1	*ATR*	Autosomal Recessive	enamel hypoplasia; malocclusion
#211900. TUMORAL CALCINOSIS, HYPERPHOSPHATEMIC, FAMILIAL; HFTC	*GALNT3*	Autosomal Recessive	enamel hypoplasia, dysplastic teeth, renal dysfunction, vascular calcification
#212750. CELIAC DISEASE; CD	*CELIAC1*	Autosomal Recessive	enamel hypoplasia
#216400. COCKAYNE SYNDROME, TYPE A; CSA	*ERCC8*	Autosomal Recessive	enamel hypoplasia/dental caries
#217080. JALILI SYNDROME	*CNNM4*	Autosomal Recessive	cone-rod dystrophy and amelogenesis imperfecta
#218040. COSTELLO SYNDROME; CSTLO	*HRAS*	Autosomal Recessive	enamel hypomineralization, delayed dental development, nail abnormalities, CNS involvement
#225410. EHLERS–DANLOS SYNDROME, TYPE VII, AUTOSOMAL RECESSIVE	*ADAMTS2*	Autosomal Recessive	excises the N-propeptide of type I and type II procollagens. NPI enzyme is a metalloproteinase
#225500. ELLIS–VAN CREVELD SYNDROME; EVC	*EVC1; EVC2*	Autosomal Recessive	enamel hypoplasia/conical shaped teeth/premature eruption of primary teeth
#226650. EPIDERMOLYSIS BULLOSA, JUNCTIONAL, NON-HERLITZ TYPE	*COL17A1; LAMA3; LAMB3; LAMC2; ITGB4*	Autosomal Recessive	enamel hypoplasia, skin/mucosal fragility, nail dystrophy
#226700, 619817. EPIDERMOLYSIS BULLOSA, JUNCTIONAL, HERLITZ TYPE	*LAMA3; LAMB3; LAMC3*	Autosomal Recessive	enamel hypoplasia, skin/mucosal fragility, nail dystrophy
#226730, # 619816. EPIDERMOLYSIS BULLOSA JUNCTIONALIS WITH PYLORIC ATRESIA	*ITGB4; ITGA6*	Autosomal Recessive	enamel hypoplasia, skin/mucosal fragility, nail dystrophy
#226750. KOHLSCHUTTER–TONZ SYNDROME	*ROGDI*	Autosomal Recessive	hypomineralized enamel, CNS involvement
#233400. PERRAULT SYNDROME; PRLTS	*HSD17B4*	Autosomal Recessive	No enamel defect mentioned in abstract
#234580, 616617. HEIMLER SYNDROME; HMLR1	*PEX1, PEX6*	Autosomal Recessive	Enamel hypoplasia, nail defects, sensorineural hearing loss
#240300. AUTOIMMUNE POLYENDOCRINE SYNDROME, TYPE I; APS1	*AIRE*	Autosomal Dominant/Recessive	enamel hypoplasia, ocular involvement, alopecia, hypoparathyroidism, pituitary defects
#241510. HYPOPHOSPHATASIA, CHILDHOOD	*ALPL*	Autosomal Recessive	enamel hypoplasia/odontohypophosphatasia
#244460. KENNY–CAFFEY SYNDROME, TYPE 1; KCS1	*TBCE*	Autosomal Recessive	enamel hypoplasia, hypodontia
#245660. JUNCTIONAL EPIDERMOLYSIS BULLOSA LARYNGOONYCHOCUTANEOUS SYNDROME; LOCS	*LAMA3*	Autosomal Recessive	hypoplastic enamel, skin blistering, corneal scarring, laryngeal stenosis, nail dystrophy
#248190 HYPOMAGNESEMIA 5, RENAL, WITH OR WITHOUT OCULAR INVOLVEMENT; HOMG5	*CLDN19*	Autosomal Recessive	enamel hypoplasia, hypomineralization, discolored teeth, renal calcium wasting, eye abnormalities
#253000. MORQUIO SYNDROME, TYPE IVA	*GALNS*	Autosomal Recessive	enamel hypoplasia, spaced dentition, short stature
#253010. MORQUIO SYNDROME, TYPE IVB	*GLB1*	Autosomal Recessive	enamel hypoplasia, spaced dentition, short stature
#257850. OCULODENTODIGITAL DYSPLASIA, AUTOSOMAL RECESSIVE	*GJA1*	Autosomal Recessive	hypoplastic teeth, delayed tooth eruption, short stature, micrognathia
#259775. RAINE SYNDROME; RNS	*FAM20C*	Autosomal Recessive	enamel dysplasia, small teeth (variable) cleft palate, low circulating phosphate
#264700. VITAMIN D HYDROXYLATION-DEFICIENT RICKETS, TYPE 1A	*CYP27B1*	Autosomal Recessive	enamel hypoplasia, delayed tooth eruption, bone pain/fracture
#270200. SJOGREN–LARSSON SYNDROME; SLS	*ALDH3A2*	Autosomal Recessive	enamel hypoplasia, kyphosis, ichthyosis
#272460. SPONDYLOCARPOTARSAL SYNOSTOSIS SYNDROME; SCT	*FLNB*	Autosomal Recessive	enamel hypoplasia, failure eruption permanent teeth, short stature, aortic aneurysm
#277440. VITAMIN D-DEPENDENT RICKETS, TYPE 2A; VDDR2A	*VDR*	Autosomal Recessive	enamel hypoplasia, dental caries, delayed tooth eruption, poor growth, bone pain/fractures
#600373. CODAS SYNDROME	*LONP1*	Autosomal Recessive	enamel dysplasia, short stature, joint hypermobility, CNS involvement
#601216 DENTAL ANOMALIES AND SHORT STATURE; DASS	*LTBP3*	Autosomal Recessive	enamel hypoplasia, failure of tooth eruption, short stature, aortic aneurysm
#604278. RENAL TUBULAR ACIDOSIS, PROXIMAL, WITH OCULAR ABNORMALITIES AND MENTAL RETARDATION	*SLC4A4*	Autosomal Recessive	enamel hypoplasia, renal bicarbonate wasting, cataract, CNS involvement
#604292. ECTRODACTYLY, ECTODERMAL DYSPLASIA, AND CLEFT LIP/PALATE SYNDROME 3; EEC3	*TP63*	Autosomal Recessive	enamel hypoplasia, hypodontia, cleft lip/palate, blepharophimosis
#607626. ICHTHYOSIS, LEUKOCYTE VACUOLES, ALOPECIA, AND SCLEROSING CHOLANGITIS; ILVASC	*CLDN1*	Autosomal Recessive	enamel hypoplasia, hypodontia, hypotrichosis
#610965. XFE PROGEROID SYNDROME; XFEPS	*ERCC4*	Autosomal Recessive	enamel dysplasia, hypodontia, short stature, renal insufficiency
#611174. HAMAMY SYNDROME, HMMS	*IRX5*	Autosomal Recessive	enamel hypoplasia, hypodontia, hearing loss, myopia, cardiac
#612463. PSEUDOPSEUDOHYPOPARATHYROIDISM; PPHP	*GNAS*	Autosomal Recessive	enamel hypoplasia, delayed tooth eruption short stature, cataract
#612782. IMMUNODEFICIENCY 9; IMD9	*ORAI1*	Autosomal Recessive	enamel hypomineralization, impaired T cell proliferation/activation, nail dysplasia, hypohidrosis
#612783. IMMUNODEFICIENCY 10; IMD10	*STIM1*	Autosomal Recessive	enamel hypomineralization, impaired T cell proliferation/activation, nail dysplasia
#612921. THREE M SYNDROME 2; 3M2	*OBSL1*	Autosomal Recessive	enamel hypomineralization, delayed dental eruption, short stature, frontal bossing
#613102. HYPOTRICHOSIS AND RECURRENT SKIN VESICLES; HYPTSV	*DSC3*	Autosomal Recessive	enamel hypoplasia, thin nails, sparse hair, skin vesicles
#613254. TUBEROUS SCLEROSIS 2; TSC2	*TSC2*	Autosomal Recessive	Pitted enamel, gingival fibroma, renal cysts, CNS involvement
#613573. ECTODERMAL DYSPLASIA-SYNDACTYLY SYNDROME 1; EDSS1	*NECTIN 4*	Autosomal Recessive	enamel hypoplasia, hypodontia, cutaneous syndactyly, sparse hair, nail abnormalities
#614564. CUTANEOUS TELANGIECTASIA AND CANCER SYNDROME, FAMILIAL; FCTCS	*ATR*	Autosomal Recessive	thin discolored enamel, dental caries, oropharyngeal cancer
#615328. SHAHEEN SYNDROME; SHN	*COG6*	Autosomal Recessive	enamel hypoplasia, dental caries, CNS involvement, palmar/plantar hyperkeratosis
#615905. DEVELOPMENTAL AND EPILEPTIC ENCEPHALOPATHY 25 WITH AMELOGENESIS IMPERFECTA; DEE25	*SLC13A5*	Autosomal Recessive	enamel hypoplasia, discolored teeth, CNS involvement, seizures
#616029. ECTODERMAL DYSPLASIA/SHORT STATURE SYNDROME; ECTDS	*GRHL2*	Autosomal Recessive	enamel hypoplasia, hypodontia, dystrophic/absent nails, short stature. focal hyperkeratosis hands/feet
#618092. INTELLECTUAL DEVELOPMENTAL DISORDER WITH SPEECH DELAY, DYSMORPHIC FACIES, AND T-CELL ABNORMALITIES; IDDSFTA	*BCL11B*	Autosomal Recessive	enamel hypoplasia, hypodontia, CNS involvement, immunological manifestations
#618349. GALLOWAY–MOWAT SYNDROME 8; GAMOS8	*NUP133*	Autosomal Recessive	enamel hypoplasia, nephrotic syndrome, CNS involvement
#618363. SHORT STATURE, AMELOGENESIS IMPERFECTA, AND SKELETAL DYSPLASIA WITH SCOLIOSIS; SSASKS	*SLC10A7*	Autosomal Recessive	enamel hypomineralization, cleft lip/palate, micrognathia. short stature, short extremities
#618440. OCULOSKELETODENTAL SYNDROME; OCSKD	*PIK3C2A*	Autosomal Recessive	enamel hypoplasia, short stature, CNS involvement, hypodontia, delayed skeletal development
#619184. SHORT STATURE, FACIAL DYSMORPHISM, AND SKELETAL ANOMALIES WITH OR WITHOUT CARDIAC ANOMALIES 2; SSFSC2	*SCUBE3*	Autosomal Recessive	enamel dysplasia, hypodontia, growth abnormalities, microcephaly
#619293. BLEPHAROPHIMOSIS-IMPAIRED INTELLECTUAL DEVELOPMENT SYNDROME; BIS	*SMARCA2*	Autosomal Recessive	enamel defects, blepharophimosis, CNS involvement
#619787. EPIDERMOLYSIS BULLOSA, JUNCTIONAL 4, INTERMEDIATE; JEB4	*COL17A1*	Autosomal Recessive	enamel hypoplasia, skin/mucosal fragility, nail dystrophy
#619489. SHORT STATURE, DAUBER–ARGENTE TYPE; SSDA	*PAPPA2*	Autosomal Recessive	enamel hypomineralization, reduced bone mineral density, growth abnormalities
**X-Linked Conditions**			
#304800. DIABETES INSIPIDUS, NEPHROGENIC, X-LINKED	*AVPR2*	X-Linked	fluorosis phenotype due to polydipsia
#305600. FOCAL DERMAL HYPOPLASIA; FDH	*PORCN*	X-Linked	enamel hypoplasia, short stature, cleft lip/palate
#307800. HYPOPHOSPHATEMIC RICKETS, X-LINKED DOMINANT	*PHEX*	X-Linked	occasional enamel hypoplasia, decreased bone density
#309000. LOWE OCULOCEREBRORENAL SYNDROME; OCRL	*OCRL1*	X-Linked	enamel hypoplasia, dental cysts, ocular involvement, renal acidosis

**Table 2 genes-14-00545-t002:** Amelogenesis imperfecta-associated molecular etiology and phenotypes.

OMIM Amelogenesis Imperfecta	Gene	Inheritance	Phenotype
# 301200. AMELOGENESIS IMPERFECTA, TYPE IE; AI1E	*AMELX*	X-linked	enamel hypoplasia/hypomineralized depending on mutation and protein effect
% 301201. AMELOGENESIS IMPERFECTA, HYPOPLASTIC /HYPOMATURATION, X-LINKED 2	*Xq22-q28*	X-linked	enamel hypoplasia, hypomineralized
#104500. AMELOGENESIS IMPERFECTA, TYPE IB; AI1B	*ENAM*	Autosomal Dominant	enamel hypoplasiac—localized or generalized
#204650. AMELOGENESIS IMPERFECTA, TYPE IC; AI1C	*ENAM*	Autosomal Recessive	enamel hypoplasia—localized or generalized
#104530. AMELOGENESIS IMPERFECTA, HYPOPLASTIC TYPE IA, AI1A	*LAMB3*	Autosomal Dominant	enamel hypoplasia—thin, pitted, grooved taurodontism
# 617607. AMELOGENESIS IMPERFECTA, HYPOMATURATION TYPE	*AMTN*	Autosomal Dominant	enamel hypomineralized
#130900. AMELOGENESIS IMPERFECTA, TYPE III; AI3	*FAM83H*	Autosomal Dominant	enamel hypomineralized—localized or generalized
# 620104. AMELOGENESIS IMPERFECTA, TYPE IK; AI1K	*SP6*	Autosomal Dominant	enamel hypoplasia—generalized
#616270. AMELOGENESIS IMPERFECTA, HYPOPLASTIC TYPE, IF, AI1F	*AMBN*	Autosomal Recessive	enamel hypoplasia—generalized
#204700. AMELOGENESIS IMPERFECTA, HYPOMATURATION TYPE, IIA1; AI2A1	*KLK4*	Autosomal Recessive	enamel hypomineralized, orange brown color
#612529. AMELOGENESIS IMPERFECTA, HYPOMATURATION TYPE, IIA2; AI2A2	*MMP20*	Autosomal Recessive	enamel hypomineralized, orange brown color, enamel loss
#613211. AMELOGENESIS IMPERFECTA, HYPOMATURATION TYPE, IIA3; AI2A3	*WDR72*	Autosomal Recessive	enamel hypomineralized creamier/opaque enamel upon eruption, enamel loss
#614832. AMELOGENESIS IMPERFECTA, HYPOMATURALTION TYPE, IIA4; AI2A4	*ODAPH, (C4ORF26)*	Autosomal Recessive	enamel hypomineralized enamel fractures readily, hypersensitivity, openbite
#615887. AMELOGENESIS IMPERFECTA, HYPOMATURATION TYPE, IIA5, AI2A5	*SLC24A4*	Autosomal Recessive	enamel hypomineralized—mottled appearance
# 618386. AMELOGENESIS IMPERFECTA, TYPE IIIC; AI3C	*RELT*	Autosomal Recessive	enamel hypomineralized—rough yellow color, enamel loss
# 616221. AMELOGENESIS IMPERFECTA TYPE IH	*ITGB6*	Autosomal Recessive	enamel hypoplasia—hypomineralized
#617297. AMELOGENESIS IMPERFECTA TYPE IJ	*ACP4*	Autosomal Recessive	enamel hypoplasia—generalized
#617217. AMELOGENESIS IMPERFECTA TYPE AI26A	*GPR68*	Autosomal Recessive	enamel hypomineralized—discolored enamel

**Table 3 genes-14-00545-t003:** Environmental etiologies of developmental defects of enamel.

DDE Associated Risk Factors	Enamel Phenotype
Xenobiotic	
Fluoride	Hypomineralized—mottled to enamel fracturing
Alcohol	Hypomineralized/hypoplasia
Chemotherapy	Hypoplasia/hypomineralized
Radiotherapy	Hypoplasia/hypomineralized
Tobacco	Hypomineralized
Antibiotics (e.g., tetracyline)	Hypoplasia, hypomineralized—phenotype varies with different antibiotics
**Infection**	
Congenital syphilis	Hypoplasia
Cytomegalovirus	Hypoplasia/hypomineralized
Congenital Rubella	Hypoplasia
Fever	Hypoplasia/hypomineralized
Respiratory tract infection	Hypomineralized/hypoplasia
Ear Infection	Hypomineralized/hypoplasia
**Hypoxia**	
Neonatal Hypoxia	Hypomineralized/hypoplasia
Trauma	Hypoplasia/hypomineralized
Metabolic Stress	
Low Birth weight	Hypoplasia/hypomineralized
Starvation	Hypoplasia/hypomineralized
Hypocalcemia	Hypomineralized/hypoplasia
Vitamin D deficiency	Hypomineralized/hypoplasia

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
