# Peer review of "Enamel Phenotypes: Genetic and Environmental Determinants"

_genes, 2023, doi:10.3390/genes14030545_

Round 1

Reviewer 1 Report

1.     In order to clarify the illustration of the review’s knowledge, the reviewer strongly suggested adding the schematic representation of part of "Normal Enamel Development" and "Gene and Environment Interactions Influencing Enamel Development."

2.     The genetic variation for molar-incisor hypomineralization (MIH), the prevalence of which varies from 3% to 40% and is a worldwide problem that affects around 13% of the world’s children, should be mentioned as a separate and important part from your review. The reviewer suggests that the genetic aetiology that may contribute to MIH should be reviewed. For example, genes such as enamelin (ENAM), tuftelin interacting protein 11 (TFIP11), and tuftelin 1 (TUFT1) appear to be associated with the development of MIH [1].

Reference:

1. Jeremias F, Pierri RA, Souza JF, Fragelli CM, Restrepo M, Finoti LS, et al. Family-Based Genetic Association for Molar-Incisor Hypomineralization. Caries Res. 2016;50(3):310-8. doi: 10.1159/000445726.

Author Response

Reviewer 1

  1. In order to clarify the illustration of the review’s knowledge, the reviewer strongly suggested adding the schematic representation of part of "Normal Enamel Development" and "Gene and Environment Interactions Influencing Enamel Development."

Response:

 There are many review articles with schematics of normal enamel development and those with genes involved and many of these are cited in the manuscript.  If the publishers would like to develop a schematic for these topics I would be happy to help but do not have access to art programs or expertise to develop these.

  1. The genetic variation for molar-incisor hypomineralization (MIH), the prevalence of which varies from 3% to 40% and is a worldwide problem that affects around 13% of the world’s children, should be mentioned as a separate and important part from your review. The reviewer suggests that the genetic aetiology that may contribute to MIH should be reviewed. For example, genes such as enamelin (ENAM), tuftelin interacting protein 11 (TFIP11), and tuftelin 1 (TUFT1) appear to be associated with the development of MIH [1].

 Response:

The topic of MIH is discussed under environmentally induced enamel defects and the world average prevalence of 13% is cited as this is clearly a global health issue.  The environmental implications of childhood illness and maternal illness are discussed.  Studies suggesting genetic influence are presented including the one suggested were already included in the manuscript.  A more recent manuscript on gene-environment interaction had also been included. PLoS One 2021;16(1):e0241898

Reference:

1. Jeremias F, Pierri RA, Souza JF, Fragelli CM, Restrepo M, Finoti LS, et al. Family-Based Genetic Association 

Reviewer 2 Report

Dear author

the review paper is summarising the recent advances in the tooth enamel field. The classifications proposed are valid and can help researches and dentists dealing with pathologic enamel structures.

I suggest to use references in most parts of the text, that could bring additional information to the readers. 

The term epigenetic should be adopted (instead of the environmental). This has been indicated by yellow in the file. 

Please, provide a small chapter about studies in mice, since this is an important step bringing important information about enamel defects in humans.

It is suggested a schematic representation of functional ameloblasts and the most important genetic and epigenetic factors involved in enamel pathology.

Author Response

Reviwer 2

the review paper is summarising the recent advances in the tooth enamel field. The classifications proposed are valid and can help researches and dentists dealing with pathologic enamel structures.

Response:

The title was not changed as normal enamel is a phenotype and not a defect and both normal and abnormal are discussed.  Phenotype is what is observed including deep phenotyping that would include ultrastructure, microhardness and other properties.  Epigenetics is not the same as environmental as there are environmental influences discussed that have nothing to do with genetics (e.g. trauma). Gene-environment interactions could be epigenetics but other interactions can occur. Elevated temperature, for example, does not alter the DNA but does alter gene expression of heat shock proteins.  Hypoxia could potentially have influences on gene by affecting the cell or though epigenetics by affecting the DNA and thus gene expression.  Thus I did not replace environment and gene-environment interaction with epigenetics as this is not correct.

Nation Human Genome definition: Epigenetics (also sometimes called epigenomics) is a field of study focused on changes in DNA that do not involve alterations to the underlying sequence. The DNA letters and the proteins that interact with DNA can have chemical modifications that change the degrees to which genes are turned on and off. 

I suggest to use references in most parts of the text, that could bring additional information to the readers. 

Response:

21 additional new references were added.

The term epigenetic should be adopted (instead of the environmental). This has been indicated by yellow in the file. 

Response:

See comments above:  environmental stressors can affect cells directly but in some instances exert there affect without alter DNA methylation of other attributes that affect gene expression.

Both terms are used as they have different meanings and both contribute to enamel phenotypes seen in humans. 

Please, provide a small chapter about studies in mice, since this is an important step bringing important information about enamel defects in humans.

Response:

A  paragraph highlighting the importance of animal studies has been added with multiple references.

It is suggested a schematic representation of functional ameloblasts and the most important genetic and epigenetic factors involved in enamel pathology.

Response:

Many such schematics appear in published reviews, many of which are cited in this manuscript.  If the publishers have access to artist that can develop such works and it is felt this greatly enhances the manuscript I am happy to provide my thoughts on developing the work and adding a caption.  I do not have access to artists to do this.

Round 2

Reviewer 1 Report

The reviewer's comments have been addressed, and the reviewer would agree to accept it.